# Usefulness of a Metal Artifact Reduction Algorithm in Digital Tomosynthesis Using a Combination of Hybrid Generative Adversarial Networks

**DOI:** 10.3390/diagnostics11091629

**Published:** 2021-09-06

**Authors:** Tsutomu Gomi, Rina Sakai, Hidetake Hara, Yusuke Watanabe, Shinya Mizukami

**Affiliations:** School of Allied Health Sciences, Kitasato University, Sagamihara 252-0373, Kanagawa, Japan; rinax@kitasato-u.ac.jp (R.S.); harah@kitasato-u.ac.jp (H.H.); y-nabe@kitasato-u.ac.jp (Y.W.); shinmiz@kitasato-u.ac.jp (S.M.)

**Keywords:** tomosynthesis, metal artifact reduction, generative adversarial network, arthroplasty, radiation-dose reduction

## Abstract

In this study, a novel combination of hybrid generative adversarial networks (GANs) comprising cycle-consistent GAN, pix2pix, and (mask pyramid network) MPN (CGpM-metal artifact reduction [MAR]), was developed using projection data to reduce metal artifacts and the radiation dose during digital tomosynthesis. The CGpM-MAR algorithm was compared with the conventional filtered back projection (FBP) without MAR, FBP with MAR, and convolutional neural network MAR. The MAR rates were compared using the artifact index (AI) and Gumbel distribution of the largest variation analysis using a prosthesis phantom at various radiation doses. The novel CGpM-MAR yielded an adequately effective overall performance in terms of AI. The resulting images yielded good results independently of the type of metal used in the prosthesis phantom (*p* < 0.05) and good artifact removal at 55% radiation-dose reduction. Furthermore, the CGpM-MAR represented the minimum in the model with the largest variation at 55% radiation-dose reduction. Regarding the AI and Gumbel distribution analysis, the novel CGpM-MAR yielded superior MAR when compared with the conventional reconstruction algorithms with and without MAR at 55% radiation-dose reduction and presented features most similar to the reference FBP. CGpM-MAR presents a promising method for metal artifact and radiation-dose reduction in clinical practice.

## 1. Introduction

The causes of metal artifacts are quite complicated. Depending on the shape and density of the metal objects, the appearance of metal artifacts can vary significantly. In medical applications, metal objects can include metallic orthopedic hardware (e.g., surgical pins and clips) or equipment attached to the patient’s body (e.g., biopsy needles) [1]. A metal object can produce beam hardening, partial volume, aliasing, small-angle scatter, under-range data acquisition electrons, or overflow of the dynamic range in the reconstruction process. A previous study on digital tomosynthesis (DT) proposed several artifact compensation approaches to minimizing metal artifacts [2,3,4].

Dual-energy DT (DE-DT) has recently become available. One of the inherent capabilities of DE-DT is the generation of synthesized monochromatic images obtained at different energy levels (keV) from a single data acquisition [3,4]. By generating monochromatic images at higher energy levels (e.g., 140 keV), beam-hardening artifacts can be suppressed. However, DT devices that enable dual-energy acquisition in clinical practice are rare; generally, DT acquisition is performed through single-energy acquisition. Therefore, from the viewpoint of versatility, an improved method must be developed to achieve metal artifact reduction (MAR) in polychromatic radiography.

Based on highly attenuating metal objects, nearly all X-ray photons are attenuated, and few reach the detector, which results in under-range data acquisition electrons. Combined with the electronic noise in the data acquisition system, near-zero or negative readings are often recorded in the measured signal after offset correction. The non-perfect treatment of these signals prior to the logarithmic operation will bias the projection estimation. When this occurs, image artifacts similar to photon starvation would appear. Combined with the bias and beam-hardening effects caused by metals, both shading and streaking artifacts appear in the reconstructed image. The effect of dose differences on metal artifacts related to such corrupted readings has been reported [5,6].

Many studies have been conducted to overcome metal-induced image artifacts [1,7,8,9,10,11]. For projection samples that pass through highly attenuating metal objects, the measured values become unreliable and must be replaced or significantly modified. Projection in-painting is one of the approaches to replacing erroneous data [8,9,10,11]. Generally, this approach first identifies the projection channels corrupted by the metallic objects. The next step is to replace these channel readings with estimated projection signals generated by the nonmetallic portion of the scanned object. The new projection samples allow the reconstruction of a metal-free image volume of the scanned object.

This approach is associated with several challenges as follows [12]: inconsistency between the synthetic and actual projections, loss of low-contract objects in the inpainted projection due to influence of the synthetic projection, and a degraded spatial resolution in the final reconstruction images. To meet these challenges, advanced image-processing techniques are often used. However, all inpainting types of algorithms have limitations. Using a large amount of missing information, the signal estimation process will not be able to fully restore the information, which leads to residual artifacts.

The development of deep learning in the reconstruction of medical images has led to recent advances in MAR featuring neural networks. Park et al. [13] used U-Net [14] in the sinogram region to process artifacts associated with beam hardening in polychromatic radiographic computed tomography (CT). Zhang et al. [15] suggested that a convolutional neural network (CNN) [16] generates a priority image with less artifacts to correct the metal-corrupted regions of the sinogram. While these methods have shown reasonable results in MAR, they have limited ability to process new artifacts remaining in the reconstructed CT image. Motivated by the success of deep learning in solving inappropriate inverse problems in image processing [17,18,19], researchers have most recently formulated MAR as an image restoration problem, and its resolution improves the quality of reconstructed CT images, image-to-image translation networks [20,21,22,23,24], as well as conditional generative adversarial networks (cGAN or pix2pix) [25], thereby further reducing metal artifacts [26]. GAN improves the recognition of lesions or tissues in medical images. For example, Han et al. have reported that the combination of noise-to-image and image-to-image improves the detection accuracy of brain tumors [27]. Sandfort et al. have also reported that cycle-GAN [28] improves tissue segmentation accuracy on CT images [29].

In general, the metal mask or metal trace regions are usually small and occupy a small portion of the whole image. The network input would weaken the metal trace information, owing to the down-sampling operations of the network. Therefore, we used the mask pyramid U-Net (mask pyramid network [MPN]) [30] to retain the model trace information at each layer to explicitly enable the network to extract more discriminative features to restore the missing information in the metal trace region. As GAN has been reported to be useful for noise reduction [31], including MAR, it can be useful for image quality improvement (artifact reduction) in DT.

In this paper, we present a novel projection-based cross-domain learning framework for generalizable MAR. Distinct from previous image restoration-based solutions, MAR was formulated as a deep learning- and projection-based completion task and training for a deep neural network, that is, a combination of multiple GANs (cycle, pix2pix, and MPN) to restore the unreliable projections within the metal region. The prior metal-free image would provide a good estimation of the missing projections [23]. To ease the pix2pix learning and improve the completion quality, we trained another neural network, cycle-GAN, to generate a good prior image with fewer metal artifacts and guide the pix2pix learning by standardizing the low-dose projection of the prior image. Moreover, we designed a novel mask pyramid projection learning strategy that would fully utilize the prior projection guidance to improve the continuity of projection completion, thereby alleviating the new artifacts in the reconstructed DT images. The final DT image was then reconstructed from the completed projection using the conventional filtered back projection (FBP) algorithm [32]. Compared with the previous MAR approaches for DT [2,3,4], the whole framework was trained efficiently for the complementary learning of prior processed images according to the method of each network so that the prior image generation and deep projection completion procedures can be learned in a collaborative manner and benefit from each other. Our recommended MAR algorithm (combination of hybrid GAN: cycle-GAN_pix2pix_MPN [CGpM-MAR]) is described in the Methods section.

In addition, we investigated the causal relationship between dose reduction and quality of MAR. Our findings suggest the possibility of reducing exposure dose and improving image quality using the CGpM-MAR algorithm. The developmental process and basic evaluation of the method are presented in this study.

## 2. Materials

### 2.1. Phantom Specifications

To evaluate image quality (with the implant and artificial bone introducing the artifacts and contrast, respectively), we immersed a prosthesis phantom containing an implant in the center of a polymethyl methacrylate case (custom-made product, Kyoto Kagaku Co., Tokyo, Japan) filled with water (case dimensions: φ 200 × 300 mm). Given that water is often used as a substitute for soft tissue in phantom experiments, the area of the phantom filled with water thus simulated soft tissue. The phantom was an artificial bone (orthopedic humeral model: normal anatomy; foam cortical shell; canal diameter: 9 mm; overall length: 300 mm, Pacific Research Laboratories Inc., Vashon, WA, USA). The TRIGEN Humeral Nails Proximal Straight System (Model: 38153000; titanium alloy; diameter: 8 mm; overall length: 160 mm, Smith & Nephew Orthopaedics KK Inc., Tokyo, Japan) was used. In the prosthesis phantom, we used an internal fracture fixation (intramedullary fracture fixation) to simulate a humeral proximal fracture.

### 2.2. DT System

The DT system (SonialVision Safire II, Shimadzu Co., Kyoto, Japan) contained an X-ray tube (anode, made of tungsten with rhenium and molybdenum; real filter; aluminum [1.1 mm]; additional aluminum [0.9 mm]) with a 0.4 mm focal spot and amorphous selenium (362.88 × 362.88 mm) digital flat-panel detector (detector element, 0.15 × 0.15 mm). The distances between source (focal spot)-to-isocenter and source-to-detector were 924 and 1100 mm, respectively (anti-scatter grid, focused type; grid ratio, 12:1). Tomography was performed linearly with a total acquisition time of 6.4 s (reference radiation dose: 80 kVp, 250 mA, and 20 ms/view; effective dose, in accordance with the International Commission on Radiological Protection [ICRP]: 0.69 mSv [ICRP 103]; 37% reduced radiation dose: 80 kVp, 250 mA, 12.8 ms/view, and 0.43 mSv; 55% radiation-dose reduction: 80 kVp, 250 mA, 9.6 ms/view, and 0.31 mSv; and 71% reduced radiation dose: 80 kVp, 250 mA, 6.4 ms/view, and 0.20 mSv). The acquisition angle was 40° and 74 projection images (1024 × 1024 matrix). The effective dose was calculated using Monte Carlo-based software (PCXMC version 2.0, STUK-Radiation and Nuclear Safety Authority, Helsinki, Finland) [33]. The reference radiation dose was the dose generally used in clinical practice (the clinical task was to assess the prosthesis). To produce reconstructed tomograms of the required height, we used a 512 × 512 matrix with 32 bits (single-precision floating number) per image.

## 3. Methods

### 3.1. Overview of CGpM-MAR

The novel CGpM-MAR algorithm was implemented in the case of projection-space data to reduce metal artifacts while reducing the radiation dose during DT. This method is based on a combination of multi-training networks (cycle-GAN and pix2pix) with MPN and a mask pyramid learning strategy to fully utilize prior projection guidance completion, and thus, alleviate new artifacts in a projection space involving hybrid and subjectively reconstructed MAR images.

Cycle-GAN allows translations between domains (reference and low doses) that are not fully associated and can solve problems of image quality deterioration due to low-dose acquisition. By applying another style to the image during the pix2pix translation process to get prior projection images, final MPN learning could be introduced to recover more structural and anatomically plausible information from the metallic component.

The overview of the recommended CGpM-MAR) algorithm is as follows:
Step 1[Cycle-GAN]: Translates low-dose images to reference-dose images.Step 2[Prior images for linear interpolation (LI) [8] processing and pix2pix GAN]: Generates a metal-free image by extracting the metal region and by LI interpolation. A prior image with a deep neural network (pix2pix) is generated to facilitate the projection completion.Step 3[MPN]: The mask information is obtained across the network’s encoding layers, and a mask fusion loss reduces the early saturation of adversarial training.

The flowchart shows the interrelations of cycle-GAN, pix2pix, and MPN, which form the core of CGpM-MAR (Figure 1).

### 3.2. CGpM-MAR

The variable definitions required for the outline of the algorithm are shown below:
*B*  reference-dose projection domain*A*  low-dose projection domain*a*  training samples ${\left\{a_{i}\right\}}_{i=1}^{Q}$ where a∈A*b*  training samples ${\left\{b_{j}\right\}}_{j=1}^{P}$ where b∈Ba~p(a)b~p(b) data distribution
AB  mapping  A→BBA  mapping  B→A*D_A_*  distinguish between images {a}
  and translated images BA{b}*D_B_*  discriminate between {b}nd AB{a}*D*  discriminator*G*  generator

#### 3.2.1. Cycle-GAN

The purpose of cycle-GAN is to translate low-dose projections into reference-dose projections. In the learning process, a low-dose projection can be effectively translated into a reference-dose projection using two discriminators and a generator. Cycle-GAN is used to learn mapping functions between two domains *A* and *B* that were given training samples. The network objective contains three types of terms: Adversarial loss, cyclic consistency loss, and identity loss. The training dataset included 148 projection images and each corresponding image related to the input image pair (*A* (74), *B* (74)) were randomly selected from the generated acquisition data as the training set.

Input_1 →*B* (reference)

Input_2 →*A* (objective)

The adversarial loss is given as
(1)$${ℒ}_{GAN}\left(AB,D_{B},A,B\right)=E_{b\sim p(b)}\left[\log D_{B}(b)\right]+E_{a\sim p(a)}\left[\log(1-D_{B}(AB(a))\right]$$

The cyclic consistency loss is given as
(2)a→AB(a)→BA(AB(a))≈a
(3)b→BA(b)→AB(BA(b))≈b
(4)$${ℒ}_{cyc}\left(AB,BA\right)=E_{b\sim p(b)}\left[{\left\|AB(BA(b))-b\right\|}_{1}\right]+E_{a\sim p(a)}\left[{\left\|BA(AB(a))-a\right\|}_{1}\right]$$

The identity loss is given as
(5)$${ℒ}_{identity}=E_{a\sim p(a)}\left[{\left\|BA(a)-a\right\|}_{1}\right]+E_{b\sim p(b)}\left[{\left\|AB(b)-b\right\|}_{1}\right]$$

The total loss is given as
(6)$$\begin{array}{ll} ℒ\left(AB,BA,D_{A},D_{B}\right) & ={ℒ}_{GAN}(AB,D_{B},A,B)\\ & +{ℒ}_{GAN}(BA,D_{A},B,A)\\ & +\lambda {ℒ}_{cyc}(AB,BA)\\ & +{ℒ}_{identity} \end{array}$$
where *λ* controls the relative importance of the two objectives. In this study, *λ* was set to 10 at an initial learning rate of 0.0002 [28].

cycle-GAN was developed to solve the following problem:(7)$$AB^{†},BA^{†}=\arg\underset{AB,BA}{\min}\underset{D_{A},D_{B}}{\max}ℒ\left(AB,BA,D_{A},D_{B}\right)$$

In cycle-GAN, the Adam optimization algorithm [34] was used with a batch size of 1. The architecture of the building components is shown in Appendix A (Table A1).

#### 3.2.2. Linear Interpolation

We generated a prior image with a deep neural network to facilitate the projection completion procedure [23], as the metal-free prior image would provide a good estimation for the original projection. We first applied the LI [8] to produce an initial estimation of the metal trace region and acquire the LI-corrected projection *P_temp_LI_* for the following procedures:
Step_1:extract metal from $AB^{†}\in {\left\{0,1\right\}}^{i\times j}$(*i*: high, *j*: width) → $M_{sk}$Step_2:$AB^{†}-M_{sk}$ → *P_temp_*Step_3:LI processing for *P_temp_* → *P_temp_LI_*


#### 3.2.3. Pix2pix

In this procedure, the original projection with metal artifacts was used as the input and to train a neural network to generate the prior image with fewer metal artifacts. In particular, with relatively large metal objects, the metal artifacts in the original projection would be strong, and the neural network would have difficulty reducing the metal artifacts. Therefore, besides the original projection, we also used the LI-corrected image into the prior image generation procedure [23] and used the pix2pix neural network, to refine the LI-corrected image by backpropagation learning.

*G* and *D* trained adversarially with GAN are expressed as follows:

Input_1 →$AB^{†}$ (objective)

Input_2 →*P_temp_LI_* (reference).

The training dataset included 148 projection images, and each corresponding images related to the input image pair ($AB^{†}$(74), *P_temp_LI_* (74)) were randomly selected as the training set from the cycle-GAN processed data:(8)$$\begin{array}{ll} \underset{D}{\min}{ℒ}_{GAN\_c}= & E_{AB^{†},P_{temp\_LI}}\left[\log D\left(AB^{†},P_{temp\_LI}\right)\right]\\ & +E_{AB^{†},z}\left[\log(1-D(AB^{†},G(AB^{†},z))\right] \end{array}$$
(9)$$\underset{G}{\min}{ℒ}_{GAN\_L1}=E_{AB^{†},P_{temp\_LI,}z}\left[{\left\|P_{temp\_LI}-G\left(AB^{†},z\right)\right\|}_{1}\right]$$
where *z* is the random noise vector (Gaussian noise).

The pix2pix was developed to solve the following problem:(10)$$AB^{††}=\arg\underset{G}{\min}\underset{D}{\max}{ℒ}_{GAN\_c}\left(G,D\right)+\gamma \underset{G}{\min}{ℒ}_{GAN\_L1}(G)$$
where γ controls the relative importance of the two objectives ${ℒ}_{GAN\_c}$ and ${ℒ}_{GAN\_L1}$. In this study, γ was set to 100, the initial learning rate was set to 0.0002, and the momentum parameters were set to *β*_1_ = 0.5, *β*_2_ = 0.999 [25].

In pix2pix, we used the Adam optimization algorithm [34] with a batch size of 1. The architecture of the building components is shown in Appendix A (Table A2).

#### 3.2.4. MPN

We used a novel mask pyramid projection learning strategy to fully utilize the prior projection guidance to improve the continuity of projection completion and thus alleviate the new artifacts in the reconstructed DT images. We used the MPN to retain the metal trace information in each layer explicitly so that the network can extract more discriminative features to restore missing information in the metal trace region.

Input_1 →*P_temp_* (objective)

Input_2 →$AB^{††}$ (reference)

Input_3 →$M_{sk}$ (objective)

The training dataset included 222 projection images and each corresponding related to the input image (*P_temp_* (74), $AB^{††}$(74), $M_{sk}$(74)) were randomly selected as the training set from the data processed in metal extract, LI, and pix2pix.

In computing the loss function, the network only considered the content of the metal mask. The content loss is given as follows:(11)$$\underset{G}{\min}{ℒ}_{c}=E_{P_{temp},AB^{††}}\left[{\left\|\hat{AB^{††}}-AB^{††}\right\|}_{1}\right]$$
(12)$$\mathrm{where}\;\hat{AB^{††}}=M_{sk}☉G\left(P_{temp}\right)+\left(1-M_{sk}\right)☉P_{temp}$$

The output score matrix from the discriminator was modulated by the metal mask $M_{sk}$ so that the discriminator can selectively ignore the unmasked regions. The adversarial part of the mask fusion loss is given as:(13)$$\begin{array}{ll} \underset{D}{\min}{ℒ}_{GAN\_M}= & E_{AB^{††}}\left[{\left\|S(M_{sk})☉\left(1-D\left(AB^{††}\right)\right)\right\|}^{2}\right]\\ & +E_{P_{temp}}\left[{\left\|S\left(M_{sk}\right)☉D\left(\overset{\wedge }{AB^{††}}\right)\right\|}^{2}\right] \end{array}$$
(14)$$\underset{G}{\min}{ℒ}_{GAN\_M}=E_{P_{temp}}\left[{\left\|S(M_{sk})☉\left(1-D\left(\hat{AB^{††}}\right)\right)\right\|}^{2}\right]$$
where *S* takes metal mask $M_{sk}$ as the input, and each block of *S* is coupled with an encoding block in *D*.

The total mask fusion loss is given as
(15)$$AB^{final}={ℒ}_{GAN\_M}+\eta {ℒ}_{c}$$
where η is the balance the between ${ℒ}_{GAN\_M}$ and η. In this study, η was set to 100, the initial learning rate was set to 0.0002, and the momentum parameters were set to $\beta_{1}=0.5{,}_{}^{}\beta_{2}=0.999$ [30]. 

In MPN, we used the Adam optimization algorithm [34] with a batch size of 1. The architecture of the building components are shown in Appendix A (Table A3). 

### 3.3. Evaluations

#### 3.3.1. Optimization Parameters of the Epochs

The optimizations epochs in each network (cycle-GAN, pix2pix, and MPN) were evaluated based on the mean square error (MSE) [35] and structural similarity (SSIM) [36] for the projection image (straightforward on the detector). The MSE of the identified projection image can be obtained as follows:(16)$$MSE=\frac{1}{mn}{\sum_{i=0}^{m-1}\sum_{j=0}^{n-1}\left[{ℜ}^{ref}(i,j)-{ℜ}^{low}(i,j)\right]}^{2}$$
where ${ℜ}^{ref}(i,j)$ is the (*i*, *j*)th entry of the reference-dose projection image, and ${ℜ}^{low}(i,j)$ is the (*i*, *j*)th entry of the low-dose projection image in each epoch.

The SSIM index between pixel values *i* and *j* was calculated as follows:(17)$$SSIM(i,j)={\left[l(i,j)\right]}^{\omega }\cdot {\left[c(i,j)\right]}^{\xi }\cdot {\left[s(i,j)\right]}^{\psi }$$
where 𝑙 is the luminance, 𝑐 is the contrast, and 𝑠 is the structure (ω=ξ=ψ=1.0).

The mean SSIM (MSSIM) was then used to evaluate the overall image quality as follows:(18)$$MSSIM[{ℜ}^{ref}(i,j),{ℜ}^{low}(i,j)]=\frac{1}{K}\sum_{q=1}^{K}SSIM(i_{q},j_{q})$$
where 𝑖*_q_* and 𝑗*_q_* are the image contents at the *q*th pixel, and *K* is the number of pixels in the image.

Optimization was evaluated based on the MSE and MSSIM. The lowest MSE, highest MSSIM, and epochs were selected as the optimum parameters.

#### 3.3.2. Evaluation of Image Quality

The DT system-derived real projection data were used for image reconstruction. MATLAB (version 9.7.0.1216025, MathWorks, Natick, MA, USA) was used to reconstruct and process images. The artifact index (AI) values [37] for metal artifacts containing low-frequency components in the CGpM-MAR and the conventional algorithms (FBP (reference-dose FBP image; reference FBP, low-dose FBP image; original FBP), DT-MAR [2], and CNNMAR [15] with reconstruction from the original projections) were compared to assess the decrease in metal artifacts on each in-focus plane image. A weighting factor of 0.6 was used in DT-MAR [(FBP∗0.4)+(BP∗0.6)] (*FBP*; FBP reconstructed images and *BP*; back projection imgaes) [2]. We further ascertained the metal artifacts containing high-frequency components. Gumbel distributions [3,38] are statistical models for determining the influence of high-frequency artifacts. The characteristics of the CGpM-MAR and conventional algorithms were evaluated based on the MAR. CGpM-MAR was evaluated using the optimized parameters generated based on the application image.

#### 3.3.3. AI

The AI of the identified metal artifacts was calculated as follows:(19)$$AI_{n}=\sqrt{\left|ROI_{artifact\_n}-ROI_{BG}\right|}$$
where *n* = 1, 2, …, 6, or 8 defines the formula for $ROI_{artifact\_n}$ (regions of interest (ROI)_location_1; $ROI_{artifact\_1}$, $ROI_{artifact\_2}$, …, $ROI_{artifact\_6}$, ROI_location_2; $ROI_{artifact\_1}$, $ROI_{artifact\_2}$, …, $ROI_{artifact\_8}$; Figure 2a) that represents the corresponding ROI for the relative standard deviations (SDs) of real features (metal artifacts) in the in-focus plane. $ROI_{BG}$ is the relative SD of the background in the in-focus plane. To evaluate each feature (metal artifacts) and background, the ROI was set at 3 × 9 pixels (ROI_location_1; streak artifact area) and 3 × 7 pixels (ROI_location_2; dark artifact area).

In this study, we compared the AI values of the reconstruction algorithms at different radiation doses between two groups (ROI_location_1 and ROI_location_2). The numbers of samples in the groups were 18 (ROI_location_1) and 24 (ROI_location_2). Statistical analyses were performed using IBM SPSS Statistics for Windows (version 24.0, SPSS Inc., Chicago, IL, USA). Probability (*p*) values < 0.05 were considered statistically significant.

#### 3.3.4. Statistical Model with Gumbel Distributions

The analysis method is outlined as follows: First, a rectangular window with a width of 8 pixels and a length (X-ray sweep direction) of 24 pixels was placed on each in-focus plane image to reduce metal artifacts, as shown in Figure 2b. Second, the parallel-line profiles of the pixel values at 1-pixel intervals resulted in 23 parallel-line pixel-value profiles (each sampling size: 23). Third, the pixel-value profiles were graphed, and the maximal variations between the adjacent pixel values were determined and analyzed based on the Gumbel distribution. Finally, the cumulative probability function was measured using the symmetry rank method with order statistics:(20)$$\mathrm{cumulative}\;\mathrm{probability}\;\Gamma \left(x_{\delta }\right)=\frac{\delta -0.5}{q},\left(\delta =1,\ldots_{}^{}q\right)$$
where *q* is the sampling size with order statistics.

To evaluate linearity, the Pearson correlation coefficient was determined and analyzed (*p* values < 0.01) using IBM SPSS Statistics for Windows (version 24.0, SPSS Inc., Chicago, IL, USA).

## 4. Results

### 4.1. Optimization Parameters

After measuring the MSE and SSIM of each training network at different radiation doses, the optimal epoch was selected at the lowest MSE and highest SSIM (0.43 mSv: cycle-GAN, 1600 epochs; pix2pix, 300 epochs; MPN, 330 epochs; 0.31 mSv: cycle-GAN, 1600 epochs; pix2pix, 280 epochs; MPN, 250 epochs; and 0.2 mSv: cycle-GAN, 1900 epochs; pix2pix, 250 epochs; and MPN, 400 epochs). Using the optimization verification results, each training network image was generated by setting epochs for CGpM-MAR and then evaluated and compared with those of the images obtained using conventional algorithms (Figure 3). The training was performed on a GPU (Geforce RTX 2080 Ti; 11 GB of memory, NVIDIA Co., Santa Clara, CA, USA). The total calculation time required to process the CGpM-MAR algorithm was 116.84 h (cycle-GAN [epochs 1900]; 79.63 h, pix2pix [epochs 300]; 10.05 h, MPN [epochs 400]; 27.16 h).

### 4.2. Image Quality

Figure 4 shows the reconstructed images of the prosthesis phantom acquired with the CGpM-MAR algorithm and each of the established algorithms for reconstruction with and without MAR processing at 55% reduction of the radiation dose of up to 0.31 mSv. Remarkably, the DT images produced using the CGpM-MAR algorithm showed decreased metal artifacts in the radiographic sweep direction (i.e., vertical direction), specifically in the peripheral regions of the prosthesis phantom. On the other hand, images produced with the help of DT-MAR demonstrated noise and metal artifacts. Comparison of the difference between CGpM-MAR and the conventional algorithm resulted in the smallest DT-MAR. CNNMAR showed a certain reduction in artifacts, but the streak artifacts generated from around the metal were remarkable.

Figure 5 presents the placement of the ROI in the prosthesis phantom and a plot of the AI results. CGpM-MAR yielded the lowest metal artifact characteristic values and features most similar to the reference FB,P regardless of the MAR processing status at 55% radiation-dose reduction, which is up to 0.31 mSv (ROI_location_1; 0.43 mSv: average AI ± standard error: 0.1020 ± 0.0179, 0.31 mSv: 0.1416 ± 0.0173, 0.20 mSv: 0.1651 ± 0.0142, ROI_location_2; 0.43 mSv: 0.1295 ± 0.0036, 0.31 mSv: 0.1226 ± 0.0051, and 0.20 mSv: 0.4308 ± 0.0313). For conventional imaging algorithms with and without MAR processing, metal artifact generation depended on the type of reconstruction algorithm ([reference FBP] ROI_location_1; 0.1103 ± 0.0267, ROI_location_2; 0.3021 ± 0.0356, [original FBP_0.43 mSv] ROI_location_1; 0.2155 ± 0.0082, ROI_location_2; 0.3893 ± 0.0309, [original FBP_0.31 mSv] ROI_location_1; 0.1998 ± 0.0182, ROI_location_2; 0.3850 ± 0.0239, [original FBP_0.20 mSv] ROI_location_1; 0.1924 ± 0.0165, ROI_location_2; 0.4264 ± 0.0283, [DT-MAR_0.43 mSv] ROI_location_1; 0.1496 ± 0.0246, ROI_location_2; 0.1469 ± 0.0067, [DT-MAR_0.31 mSv] ROI_location_1; 0.1579 ± 0.0305, ROI_location_2; 0.1439 ± 0.0058, [DT-MAR_0.20 mSv] ROI_location_1; 0.1578 ± 0.0297, ROI_location_2; 0.1411 ± 0.0062, [CNNMAR_0.43 mSv] ROI_location_1; 0.1493 ± 0.0191, ROI_location_2; 0.3473 ± 0.0388, [CNNMAR_0.31 mSv] ROI_location_1; 0.1768 ± 0.0273, ROI_location_2; 0.3559 ± 0.0402, [CNNMAR_0.20 mSv] ROI_location_1; 0.2142 ± 0.0165, ROI_location_2; 0.3512 ± 0.0407).

In ROI_location_1, the differences in the metal artifacts, except for CGpM-MAR compared with the conventional low-dose FBP without MAR processing, were statistically significant (*p* < 0.05; Table 1). In ROI_location_2, the differences in metal artifacts between the CGpM-MAR and low-dose with and without MAR processing reconstruction algorithms were statistically significant (*p* < 0.05; Table 2). These AI results showed that the CGpM-MAR algorithm most effectively reduced metal artifacts in areas with a large number of artifacts (ROI_locations_2). In addition, CGpM-MAR had the closest AI value to the reference FBP.

Figure 6 shows a Gumbel plot of the relationships between the largest variations and estimated cumulative probabilities. Here, the largest variations are distributed linearly (reference FBP *r* = 0.925 [*p* < 0.01], 0.43 mSv: original FBP *r* = 0.909 [*p* < 0.01]; DT-MAR *r* = 0.935 [*p* < 0.01]; CNNMAR *r* = 0.919 [*p* < 0.01]; CGpM-MAR *r* = 0.919 [*p* < 0.01], 0.31 mSv: original FBP *r* = 0.961 [*p* < 0.01]; DT-MAR *r* = 0.982 [*p* < 0.01]; CNNMAR *r* = 0.930 [*p* < 0.01]; CGpM-MAR *r* = 0.976 [*p* < 0.01], 0.20 mSv: original FBP *r* = 0.986 [*p* < 0.01]; DT-MAR *r* = 0.981 [*p* < 0.01]; CNNMAR *r* = 0.918 [*p* < 0.01]; CGpM-MAR *r* = 0.986 [*p* < 0.01]). These findings further verified the Gumbel distribution as a reasonable statistical model for describing the largest variations in each largest difference between adjacent pixel-value profiles. Furthermore, the largest variations in the Gumbel plot showed the CGpM-MAR had the minimum number of high-frequency artifacts and features most similar to the reference FBP at 55% radiation-dose reduction, which is up to 0.31 mSv. Furthermore, the original FBP and CNNMAR algorithms exhibited similar distributions at 55% radiation-dose reduction of up to 0.31 mSv. Whereas, the original FBP algorithm distribution differed at 71% radiation-dose reduction of up to 0.20 mSv. The AI results and Gumbel distribution indicate that CGpM-MAR had characteristics similar to the reference FBP at doses of up to 0.31 mSv (55% reduction).

## 5. Discussion

This study revealed that the CGpM-MAR algorithm yielded an adequate overall performance, reducing the radiation dose by 55%. The combination of multi-training network images produced using this algorithm yielded good results independently of the type of metal present in the prosthesis phantom. In addition, this algorithm successfully removed low- and high-frequency artifacts from the images. CGpM-MAR was particularly useful in reducing a large number of artifacts. Therefore, this algorithm is a promising new option for prosthetic imaging, as it generated artifact-reduced images and reduced radiation doses that were far superior to those obtained from images processed using conventional algorithms. The flexibility of CGpM-MAR in the choice of imaging parameters, which is based on the desired final images and prosthetic imaging conditions, promises increased usability.

The projection-space combination of multi-training approaches described here can be used to generate images to formulate the MAR as a deep learning algorithm for projection completion problems to improve the generalization and robustness of the framework. Since directly regressing accurate missing projection data is difficult to undertake [23], we propose to incorporate the prior projection image generation procedure and adopt a combination of multiple networks and a projection completion strategy. This method can improve the continuity of the projection values at the boundary of metal traces and alleviate the new artifacts, which are common drawbacks of projection completion-based MAR methods. Therefore, we believe that our novel CGpM-MAR could effectively reduce metal artifacts in actual practice.

The ability of CGpM-MAR to obtain MAR images and reduce the radiation dose by approximately 55% (Figure 5 and Figure 6) may be due to the benefits of the first process, cycle-GAN. Training an image-to-image translation framework requires fully associated images, which is often difficult to learn. Cycle-GAN allows the translation between domains that are not fully associated and can therefore solve this problem. Cycle-GAN has three types of losses: First, the cycle consistency loss calculates the difference between the original image and the original domain after translation into another domain and the original domain. Second, adversarial loss guarantees that the image is real. Third, identity loss preserves the quantization of the pixel space of the image. The two generators use a U-Net [14] structure, and the two discriminators have a Patch-GAN-based structure [39] for learning. By applying another style to the image during the translation process, the low-dose projection image can then be applied to the reference-dose projection image.

The MPN used as the final learning process in CGpM-MAR contributed to the MAR. The reason was that adversarial learning could be introduced into the projection to recover more structural and anatomically plausible information from the metallic domain. In addition, a new MPN has been developed, which extracts geometric information of different scales and mask fusion loss that penalizes premature saturation, making learning more robust for different shapes of metal implants, were introduced.

In DT-MAR processing without deep learning methods [2], voxels containing artifacts tended to have higher values than their neighboring artifact-free voxels, which affected the prosthetic appearance, wherein former voxels stood out against the background of the latter. Accordingly, these residual artifacts are conspicuous when images subjected to the FBP method are compared with non-artifact-reduced images. Therefore, DT-MAR processing based on the polychromatic radiographic imaging method has a limited ability to reduce metal artifacts.

The usefulness of image quality improvement for reducing noise and metal artifacts in DT using deep learning has recently been reported [4,31]. Although noise and radiation-dose reductions using deep learning in the DT of the breast are possible, no studies have reported on the reduction of radiation dose related to MAR. The radiation-dose reduction is approximately 20% in MAR without deep learning [6], thus, applying deep learning can further improve MAR and radiation-dose reduction.

Our CGpM-MAR algorithm has some limitations. First, this study was to reduce DT metal artifacts by single-energy (polychromatic) acquisition depending on the versatility of the processing. When DE-DT acquisition becomes widespread in the future, we would like to apply CGpM-MAR in monochromatic radiographic imaging. Second, because CGpM-MAR combines multiple deep learning processes, generating the final image takes time. Therefore, improvements in hardware processing are desired to speed up the process. Third, the model used for learning used a phantom. By acquiring data, such as prosthesis size and the positional relationship between prosthesis and normal structure, under various conditions according to clinical use, image quality can possibly be improved further. Moreover, further knowledge of various other structural patterns is needed in processing complicated structures.

Although the results of this study is limited to prosthesis phantom, the evaluations were performed in a state closest to actual biological composition in vivo. Furthermore, this approach will accelerate clinical application in terms of radiation-dose reduction and image quality improvement, which are issues in X-ray imaging. We believe that the CGpM-MAR algorithm will optimize the acquisition protocol in future X-ray imaging and radiation-dose reduction technology and improve the accuracy of medical images.

Recently, some works studied GAN applications with and without MAR [40,41]. These studies applied GAN technology to the reconstructed image to improve the accuracy of the tomographic image in the in-plane and longitudinal directions. Although the images processed were projection images, we predict that further improvements to the image quality (MAR) of the reconstructed image can be made by theoretically using GAN in the three-dimensional direction or in multiple slice generation from the input slice images. In addition, investigating the simultaneous reduction of GAN and metal artifacts in three-dimensional data would be an interesting future research direction.

## 6. Conclusions

This prosthesis phantom study revealed that a 55% reduction in radiation dose is feasible with our novel CGpM-MAR algorithm. Our algorithm was particularly useful in reducing dark artifacts and yielded relatively better statistical results (*P* < 0.05) in terms of metal artifact reduction than conventional reconstruction algorithms. Therefore, the CGpM-MAR algorithm can be integrated into the clinical application workflow to accelerate image acquisition and reconstruction, and reduce metal artifacts while maintaining excellent image quality in radiologic imaging with prostheses.

## Figures and Tables

**Figure 1 diagnostics-11-01629-f001:**
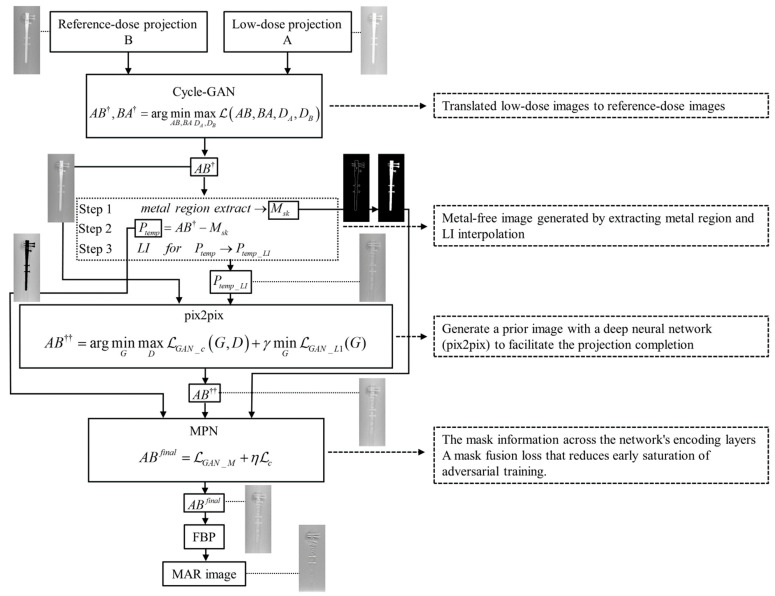
Flowchart of the combined hybrid GAN, cycle-GAN_pix2pix_MPN algorithm (CGpM-MAR). The interrelations among the algorithms are also shown.

**Figure 2 diagnostics-11-01629-f002:**
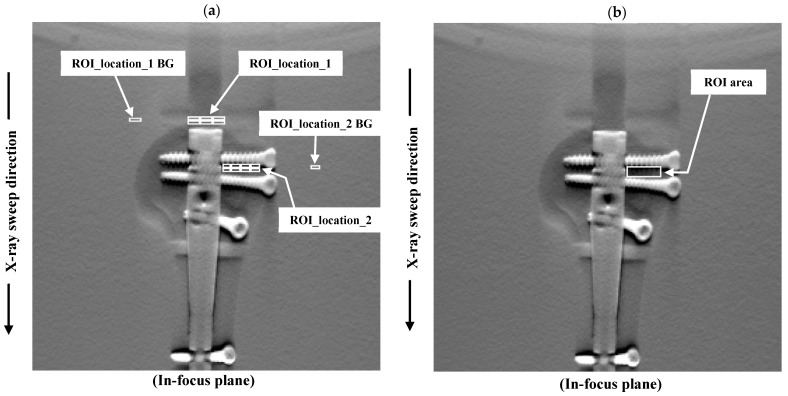
Assessment of improvements in image quality using the artifact index (AI), a statistical model with a Gumbel distribution of the selected features. The in-focus plane images shows the (**a**) artifact and background areas of the AI and (**b**) measurements and high-frequency artifact with prosthetic areas of the Gumbel analysis.

**Figure 3 diagnostics-11-01629-f003:**
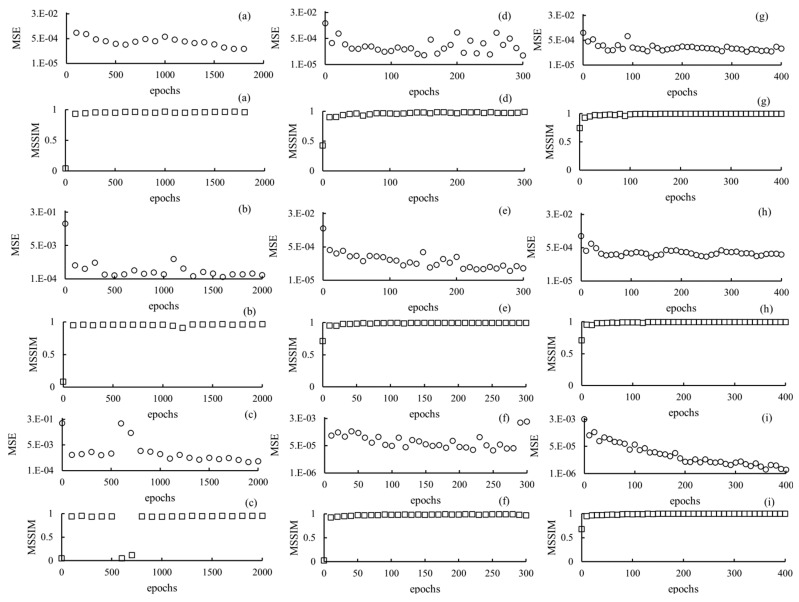
Optimization results for parameter (epochs) determination for each network algorithm at different radiation-dose levels: (**a**) cycle-GAN 0.43 mSv; (**b**) cycle-GAN 0.31 mSv; (**c**) cycle-GAN 0.20 mSv; (**d**) pix2pix2 0.43 mSv; (**e**) pix2pix 0.31 mSv; (**f**) pix2pix 0.20 mSv; (**g**) MPN 0.43 mSv; (**h**) MPN 0.31 mSv; and (**i**) MPN 0.20 mSv.

**Figure 4 diagnostics-11-01629-f004:**
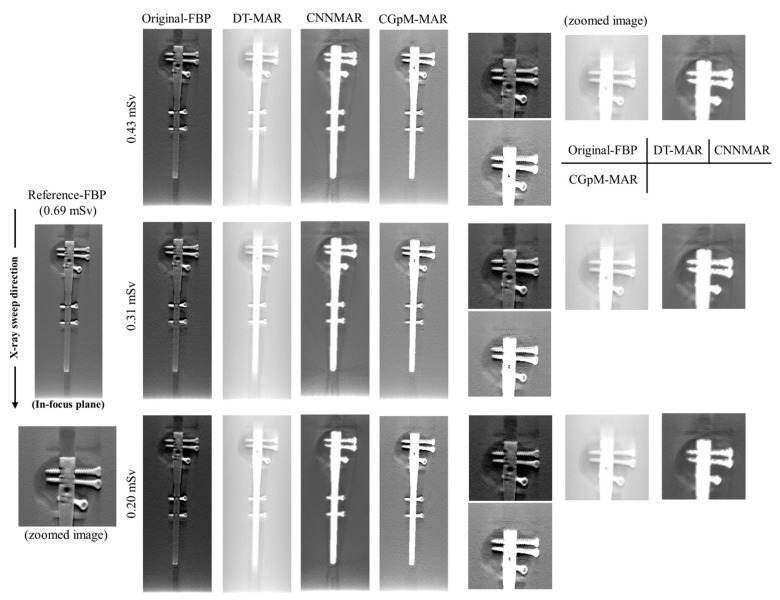
Comparisons between the combined hybrid GAN, cycle-GAN_pix2pix_MPN algorithm[CGpM-MAR, and the conventional reconstruction algorithms with and without metal artifact reduction (MAR; reference FBP [showing window: 0.15–0.44], 0.43 mSv; CGpM-MAR [0.27–0.34], original FBP [0.15–0.44], DT-MAR [0–0.72], CNNMAR [0–0.20], 0.31 mSv; CGpM-MAR [0.24–0.32], original FBP [0.15–0.44], DT-MAR [0–0.72], CNNMAR [0–0.20], 0.20 mSv; CGpM-MAR [0.05–0.22], original FBP [0.15–0.44], DT-MAR [0–0.72], CNNMAR [0–0.20]) in the in-focus plane. Metallic implants were replaced with constant values (white) after processing with CGpM-MAR and CNNMAR. The display variety of the prosthesis phantom was changed for visual comparison of the contrast and background gray levels. The X-ray source was moved along the image vertically. In the displayed areas, the artifact indexes were determined.

**Figure 5 diagnostics-11-01629-f005:**
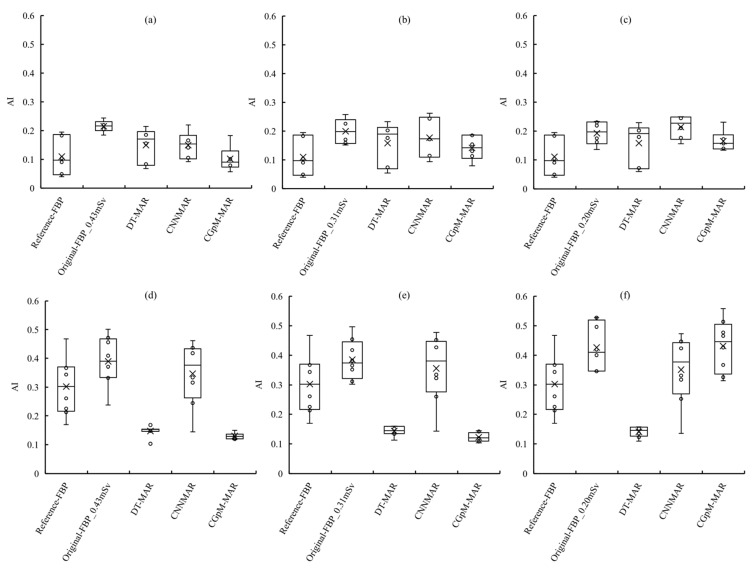
Comparisons of the artifact indexes (AIs) determined for in-focus plane images obtained using the cycle-GAN_pix2pix_MPN (CGpM-MAR) and other traditional reconstruction algorithms with and without metal artifact reduction (MAR) processing. Metal artifacts originating from the AIs of 6 (ROI_location_1) and 8 (ROI_location_2) selected metal artifact areas (features) and one background area are presented in the in-focus plane. Reference FBP, 0.69 mSv; (**a**) ROI_location_,1 0.43 mSv; (**b**) ROI_location_1, 0.31 mSv; (**c**) ROI_location_1, 0.20 mSv; (**d**) ROI_location_2, 0.43 mSv; (**e**) ROI_location_2, 0.31 mSv; and (**f**) ROI_location_2, 0.20 mSv.

**Figure 6 diagnostics-11-01629-f006:**
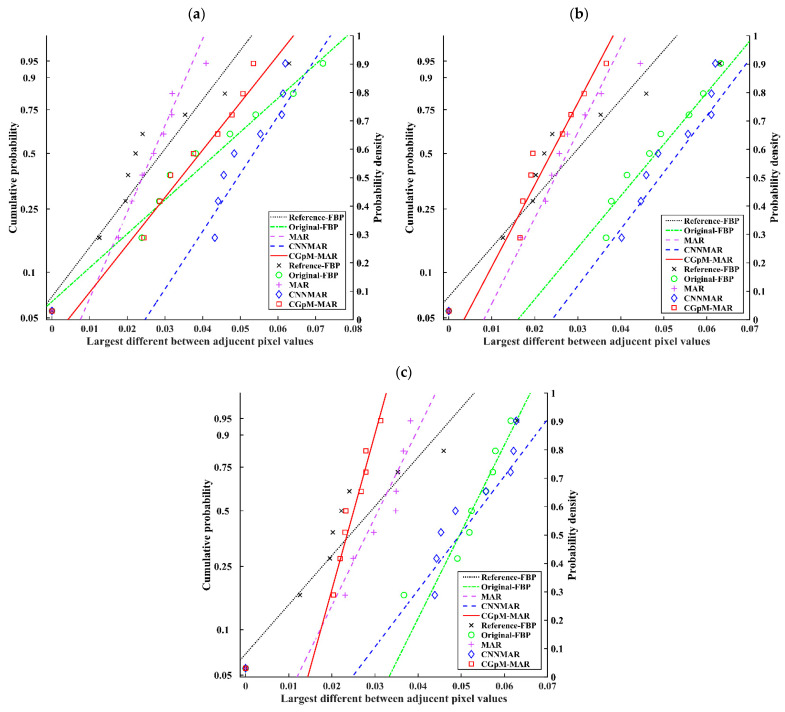
The largest variations extracted from 48 pixel-value profiles are plotted. The relatively large variation in pixel values can be attributed to the high-frequency artifacts. [(**a**) 0.20 mSv; (**b**) 0.31 mSv, and (**c**) 0.43 mSv].

**Table 1 diagnostics-11-01629-t001:** Metal artifact reduction performances of tomosynthesis reconstruction algorithms. (streak artifact area).

ROI_Location_1					
Variable	Difference	Standard Error	*p*	95% CI *	
				Lower Limit	Upper Limit
CGpM-MAR vs. Original-FBP	−0.0664	0.01717	0.002	−0.1118	−0.0211
CGpM-MAR vs. DT-MAR	−0.0190	0.01717	0.686	−0.0644	0.0263
CGpM-MAR vs. CNNMAR	−0.0441	0.01717	0.060	−0.0895	0.0013
Original FBP vs. DT-MAR	0.0474	0.01717	0.037	0.0021	0.0928
Original FBP vs. CNNMAR	0.0224	0.01717	0.565	−0.0230	0.0677
Original FBP vs. CGpM-MAR	0.0664	0.01717	0.002	0.0211	0.1118
DT-MAR vs. Original-FBP	−0.0474	0.01717	0.037	−0.0928	−0.0021
DT-MAR vs. CNNMAR	0.1389	0.0601	0.468	−0.0704	0.0203
DT-MAR vs. CGpM-MAR	0.8692	0.0537	0.686	−0.0263	0.0644
CNNMAR vs. Original-FBP	−0.0224	0.01717	0.565	−0.0677	0.0230
CNNMAR vs. DT-MAR	0.0251	0.01717	0.468	−0.0203	0.0704
CNNMAR vs. CGpM-MAR	0.0441	0.01717	0.060	−0.0013	0.0895
Source of variation	df *	sums of squares	mean square	F	*p*
Algorithm	3	0.045	0.015	5.709	0.002
Dose	2	0.010	0.005	1.827	0.170
Algorithm × Dose	6	0.017	0.003	1.086	0.381
Error	60	0.159	0.003	-	-

* CI: confidence interval; dependent variable: artificial index value. Tukey–Kramer test; *p* < 0.05 indicates a significant difference (without metal artifact reduction processing). * df: degree of freedom; dependent variable: artificial index value. Tukey–Kramer test; *p* < 0.05 indicates a significant difference (without metal artifact reduction processing).

**Table 2 diagnostics-11-01629-t002:** Metal artifact reduction performances of tomosynthesis reconstruction algorithms. (dark artifact area).

ROI_Location_2					
Variable	Difference	Standard Error	*p*	95% CI *	
				Lower Limit	Upper Limit
CGpM-MAR vs. Original-FBP	−0.1726	0.02142	0.000	−0.2288	−0.1165
CGpM-MAR vs. DT-MAR	0.0836	0.02142	0.001	0.0274	0.1397
CGpM-MAR vs. CNNMAR	−0.1238	0.02142	0.000	−0.1800	−0.0677
Original FBP vs. DT-MAR	0.2562	0.02142	0.000	0.2001	0.3124
Original-FBP vs. CNNMAR	0.0488	0.02142	0.111	−0.0074	0.1049
Original-FBP vs. CGpM-MAR	0.1726	0.02142	0.000	0.1165	0.2288
DT-MAR vs. Original-FBP	−0.2562	0.02142	0.000	−0.3124	−0.2001
DT-MAR vs. CNNMAR	−0.2074	0.02142	0.000	−0.2636	−0.1513
DT-MAR vs. CGpM-MAR	−0.0836	0.02142	0.001	−0.1397	−0.0274
CNNMAR vs. Original-FBP	−0.0488	0.02142	0.111	−0.1049	0.0074
CNNMAR vs. DT-MAR	0.2074	0.02142	0.000	0.1513	0.2636
CNNMAR vs. CGpM-MAR	0.1238	0.02142	0.000	0.0677	0.1800
Source of variation	df *	sums of squares	mean square	F	*p*
Algorithm	3	0.979	0.326	59.273	0.000
Dose	2	0.154	0.077	13.960	0.000
Algorithm × Dose	6	0.351	0.058	10.618	0.000
Error	84	0.462	0.006	-	-

* CI: confidence interval; dependent variable: artificial index value. Tukey–Kramer test; *p* < 0.05 indicates a significant difference (without metal artifact reduction processing). * df: degree of freedom; dependent variable: artificial index value. Tukey–Kramer test; *p* < 0.05 indicates a significant difference (without metal artifact reduction processing).

## Data Availability

All relevant data are within the manuscript and its non-published material files.

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
