# Peer review of "Usefulness of a Metal Artifact Reduction Algorithm in Digital Tomosynthesis Using a Combination of Hybrid Generative Adversarial Networks"

_diagnostics, 2021, doi:10.3390/diagnostics11091629_

Round 1
Reviewer 1 Report
This manuscript introduces an approach based on hybrid generative adversarial networks, named (GANs; cycle-consistent GAN_pix2pix_MPN [CGpM-metal artifact reduction [MAR]), for metal artifact reduction. In particular, projection data were analyzed to reduce metal artifacts and the radiation dose during digital tomosynthesis. The experiments were conducted on a prosthesis phantom at variable radiation doses. The evaluation and comparisons were based on the artifact index.
The results are reasonable but a generalization assessment would be beneficial for the impact of the proposed GAN-based method. The literature background should be extended, as well as the clinical application should be better clarified. Careful proofreading will be beneficial too.
My main concerns are listed in what follows.
1) Abstract: the final statement should clearly acknowledge the limitations of this phantom study.
2) The concept of "combined hybrid GAN" should be better explained since it is quite crucial for assessing the novelty of the proposed approach.
3) Section 1 (Lines 107-199): I would move these lines to Section 2.1
4) Section 2: A better separation between Materials and Methods should be considered. At the moment, they are quite mixed. Please improve the structure.
5) Regarding the latest GAN-powered applications, these highly relevant and recent articles for image reconstruction and metal artifact reduction should be introduced:
- Han, C., Rundo, L., Murao, K., Noguchi, T., Shimahara, Y., Milacski, Z. Á., ... & Satoh, S. I. (2021). MADGAN: unsupervised medical anomaly detection GAN using multiple adjacent brain MRI slice reconstruction. BMC Bioinformatics, 22(2), 31. DOI: 10.1186/s12859-020-03936-1
- Wang, J., Noble, J. H., & Dawant, B. M. (2019). Metal artifact reduction for the segmentation of the intra cochlear anatomy in CT images of the ear with 3D-conditional GANs. Medical Image Analysis, 58, 101553. DOI: 10.1016/j.media.2019.101553
Please discuss these two highly relevant publications.
6) Table 1 is very detailed and might be moved to Appendix. I would suggest a more concise description to be included in the main text as Table 1.
Moreover, the notation for the multiplication symbol should be '×' rather than 'x' (in the 'Kernel' column). Please carefully revise it throughout the manuscript.
7) Figure 1: please increase the font size when possible.
8) Figure 6: the labels and axis ticks need to be increased since they are currently unreadable.
9) Line 610: regarding "the model used for learning uses a phantom", more details on the generalization to in vivo studies should be provided.
10) Lines 616-618: The statement "Our CGpM-MAR algorithm was particularly useful for reducing a large number of artifacts and yielded relatively better results in terms of metal artifact reduction than the conventional reconstruction algorithms." is quite vague and needs more clarifications with stronger statements on the statistically significant results found.
11) Section 5: conclusive remarks need to be extended and clarified, as well as a feasible plan for future work should be provided.
Reviewer 2 Report
The paper entitled „Usefulness of a metal artifact reduction algorithm in digital tomosynthesis using a combination of hybrid generative adversarial networks” by Gomi et al. deals with a novel combination of hybrid generative adversarial networks artifact reduction in order reduce metal artifacts and the radiation dose during digital tomosynthesis.
The metal artifact reduction rates were compared using the artifact index and Gumbel distribution.
The results show that the algorithm CGpM-MAR was useful for reducing metal artifacts and radiation dose.
There are several minor concerns which should be addressed by the authors.
- The quality of all figures should be improved.
- The Related work section is missed.
- It is unclear from the manuscript how many images, from each dataset, were considered? Also, information regarding the image resolution should be useful.
- Also, it is unclear from the manuscript if the images were segmented in regions of interest, in figure 1 is mentioned the “material region extract”.
- The authors did not specify the training time/time consuming for CGpM-MAR.
- #lines 170 and 171, please to define a and b variables.
- #line 190 Please double check the number of closed brackets (eq.1).
- The conclusions supports the results, please to complete this section with and future research.
Round 2
Reviewer 1 Report
The Authors have properly addressed the comments from the previous Revision Round, by providing comprehensive responses. The contributions and conclusive remarks have been extended, as well as the quality of the Figures has been enhanced. However, the literature background might be further improved.
My last suggestions can be found in what follows.
* Section 1: When introducing Cycle-GAN, relevant applications of image-to-image GANs to medical imaging should be discussed, such as in:
- Han C., et al., (2019) Combining Noise-to-Image and Image-to-Image GANs: Brain MR Image Augmentation for Tumor Detection. IEEE Access, 7, 156966-156977, 2019. DOI: 10.1109/ACCESS.2019.2947606
- Sandfort V., et al. (2019) Data augmentation using generative adversarial networks (CycleGAN) to improve generalizability in CT segmentation tasks. Scientific Reports, 9(1), 1-9. DOI: 10.1038/s41598-019-52737-x].
Please consider introducing this important aspect.
* Section 3: please improve the formatting and the notation of the mathematical equations.
* A final proofread would be beneficial.
